# Effect of Sourdough and Whey Protein Addition on the Technological and Nutritive Characteristics of Sponge Cake

**DOI:** 10.3390/foods11141992

**Published:** 2022-07-06

**Authors:** Nikola Maravić, Dubravka Škrobot, Tamara Dapčević-Hadnađev, Biljana Pajin, Jelena Tomić, Miroslav Hadnađev

**Affiliations:** 1Faculty of Technology, University of Novi Sad, Bul. Cara Lazara 1, 21000 Novi Sad, Serbia; pajinb@tf.uns.ac.rs; 2Institute of Food Technology, University of Novi Sad, Bul. Cara Lazara 1, 21000 Novi Sad, Serbia; dubravka.skrobot@fins.uns.ac.rs (D.Š.); tamara.dapcevic@fins.uns.ac.rs (T.D.-H.); jelena.tomic@fins.uns.ac.rs (J.T.); miroslav.hadnadjev@fins.uns.ac.rs (M.H.)

**Keywords:** sourdough, whey proteins, sponge cake, quality assessment

## Abstract

Whey protein and sourdough ferment were used in different combinations to prepare functional sponge cakes, and their mutual influence on batter rheological behaviour as well as product physico–chemical, textural, colour and sensory properties were evaluated. All samples containing whey protein concentrate could bear the nutrition claim ‘a source of protein’. The substitution of wheat flour with whey protein significantly influenced batter viscoelastic behaviour, lowered cake-specific volume, increased product hardness, chewiness, gumminess, and browning index and modified its sensory characteristics. The incorporation of sourdough in protein-enriched sponge cakes improved product-specific volume and appearance compared to a protein-containing sample without sourdough. Although sourdough addition has less of a deteriorating effect on sponge cake rheological and textural properties, when combined with whey protein, it led to a significant reduction in batter elasticity and an increase in product hardness. It was also shown that spontaneously fermented sourdough cannot act as the only leavening agent in sponge cake production. In general, the results of this study have shown that sourdough addition can contribute to improvement in protein-enriched sponge cake quality and that further investigations are necessary in terms of different sourdough and flour type incorporation to minimize the negative effects of protein addition.

## 1. Introduction

Nowadays, food products undergo too many formulation and processing changes to fulfil modern world demands. These changes involve improving nutritional value, reducing, or fully replacing calorie-laden ingredients, and improving the taste and overall characteristics of the original product [1,2]. Although they cannot be classified into a group of so-called ‘healthy’ foods, sponge cakes, as a common and preferred bakery speciality, can be a promising matrix for the implementation of various functional ingredients. As it is perceived as a high-calorie and nutritiously compromised food, this type of product offers different pathways of modification in order to improve its various aspects of quality. These pathways include the fortification of sponge cakes or substitution of nutritionally less desirable ingredients with nutrient-rich ingredients to produce health-enhancing or functional sponge cakes, such as high-protein, high-fibre, low-fat and low-sugar products [3,4,5,6,7]. In addition to the undeniable improvement in nutritional quality, the incorporation of new ingredients in bakery products leads to gluten dilution, which represents a challenging issue for food technologists, as gluten plays a key role in providing the desired texture, sensory characteristics, and overall quality. There are different studies investigating the incorporation of whey protein [8], pea protein [9] or soy protein [10] in sponge cakes and similar products as egg substitutes or product fortifying agents. Most of the studies revealed product quality deterioration with an increase in protein concentration in terms of taste, colour, volume, texture and overall acceptability of the bakery product [10,11]. Therefore, protein addition to bakery products is often accompanied by structure-building or improving agents such as cross-linking enzymes or additives [12,13]. Among all tested proteins, whey proteins in sponge cake formulations gain special attention in the scientific domain. Apart from being highly nutritious, exceptional functional properties such as water-binding capacity, foaming, emulsifying and gel formation ability have made whey protein stand out as a commonly used functional ingredient [14,15].

Within the framework of improving the quality of food products, different studies have highlighted the necessity to apply some of the novel or modified food processing technologies. A trend that is increasingly attracting the attention of both the bakery industry as well as consumers is the application of sourdough technology as an approach for modification of flour functionality towards improvement in cereal-based products quality. Sourdough fermentation is a specific process where lactic acid bacteria and yeast create a complex consortium which uses the metabolism of both, as well as their mutual impact, to transform ingredients. Nutrients and bioactive compounds are thus subjected to numerous changes providing several benefits to final products in terms of nutritive, health, microbial and sensory aspects [16,17,18,19]. Namely, many scientific studies have confirmed the benefits of sourdough implementation in bakery products, such as the improvement of the flavour and texture and prolongation of the final product shelf life by decreasing the rate of firmness and starch retro-gradation. Moreover, some studies emphasise the potential of this approach to reduce anti-nutrients and enable fortification of cereal-based products with fibre, as well as decrease glycaemic index and improve protein digestibility [18,20]. It is worth noting that most of the studies on this topic are focused on sourdough bread. On the other hand, there is a certain number of review reports that, regarding the future trends, point out that the focus should be on the implementation of sourdough technology in cereal products other than bread [21,22]. According to available literature data, there are some efforts in the development of bakery products with sourdough addition such as pasta, cookies, biscuits and sponge cakes [4,23,24]. However, considering that sourdough is a very complex biological system and that it is difficult to predict its impact on various quality aspects of the final product, there is a need for further research in this area.

Therefore, the aim of this study was to evaluate the effect of whey protein and spontaneously fermented whole wheat sourdough on sponge cake quality. Contrary to the other studies which monitored the effects of egg white protein replacement with whey protein concentrate on the physical, structural, and sensory properties of sponge cake, the present research gives an insight into changes that occur when wheat flour is substituted with whey protein concentrate. According to our knowledge application of sourdough fermentation and whey protein in sponge cake formulation is mostly examined separately and there is a lack of published data using sourdough technology to increase the quality of protein-enriched sponge cake. In order to determine the effects of combined approaches, several cake formulations were created. To estimate the quality of final products, rheological properties, chemical composition, physical properties, colour, textural analysis, as well as sensory evaluation were conducted.

## 2. Materials and Methods

### 2.1. Materials

Wholemeal and refined wheat flours were obtained by Danubius d.o.o. (Novi Sad, Serbia). Whey protein concentrate (77% proteins, 7.3% fats and 8.5% of carbohydrates, all expressed on dry matter content) was purchased from Olimp Laboratories Sp.z.o.o. (Debica, Poland), while sucrose (Fabrika šećera “CRVENKA”, Crvenka, Serbia), eggs (Jaje Produkt doo, Čenej, Serbia) and baking powder (rising agents—sodium hydrogen carbonate, disodium phosphate; corn starch) (AD Aleva, Novi Kneževac, Serbia) were obtained from local market. Deionized water was used in all the experiments. Chemicals used in the analysis of sponge cakes were of analytical grade and were purchased from Merck (Darmstadt, Germany).

### 2.2. Preparation of Sponge Cake

Wholewheat flour was used for creating sourdough starter. Spontaneously fermented sourdough was prepared through backslopping (every 24 h, 5 days) under laboratory conditions (temperature of 25 °C, dough yield of 200, sourdough:flour:water = 1:2:2).

Six sponge cake formulations were prepared according to Table 1. Different amounts of wheat flour, whey protein, sourdough and baking powder were used in sponge cake formulations in order to determine their individual and combined effects on technological and nutritional quality parameters of sponge cake samples. Samples contained only wheat flour (Control), wheat flour and whey protein (P_20_), wheat flour and sourdough (S_30_) and wheat flour, sourdough, and whey protein at different ratios (S_20_P_20_ and S_30_P_20_). Furthermore, an additional sample with sourdough (S_30_P_20_BP_0_) was prepared using all ingredients, except baking powder. Mature and stable wholegrain wheat flour starter was used as sourdough in sponge cake formulations. In sourdough containing sponge cakes the amount of flour and water was reduced for the ones already present in added sourdough.

Fresh eggs were separated into albumen (egg white) and yolk. Subsequently, 52.5 g of egg albumen was whipped with 20 g of sugar in a mixer (Gorenje, Slovenia) for 1 min at mixing speed 5. Moreover, an additional amount of sugar (65 g) and egg yolk (32.5 g) was also mixed for 1 min at speed 3 and added to previously prepared mixture. Subsequently, the appropriate amount of water was added and mixed for 1 min at speed 1. Finally, the powder ingredients (wheat flour, whey protein and baking powder) were homogenised and incorporated into previously prepared mixture by mixing for 1 min at speed 1. At the end, mature sourdough was added and gently mixed for 30 s. The obtained blends were poured into silicone moulds (60-mm diameter), 30 g each per sample. Afterwards, the poured batter was baked in a conventional oven at 200 °C for 11 min and 30 s. Two batches of each sample were prepared.

Rheological properties of batter were examined, while physical characteristics, chemical composition, colour, textural properties, and sensory evaluation were determined on baked sponge cake.

### 2.3. Batter Rheology

To assess the rheological properties of batter, a HAAKE MARS Rheometer (Thermo Scientific, Karlsruhe, Germany) was used. Dynamic oscillatory measurements were performed using serrated parallel plate geometry measuring system (35-mm diameter, 1-millimetre gap) at constant temperature of 25 °C maintained by Peltier system. Mechanical spectra (frequency sweeps) were measured in the range 0.1–10 Hz at 1 Pa stress, which was within previously determined linear viscoelastic range. Three replicates were performed for each analysis.

### 2.4. Chemical Composition of Sponge Cake

Methods of the Association Official of Analytical Chemists (AOAC, 2000) were used to determine chemical composition of prepared sponge cakes. Moisture, fat, protein, ash and total fibre content were analysed (Methods No. 925.10, 920.85, 920.87, 923.03, and 985.29, respectively). Total carbohydrates were calculated by difference following Equation (1) and total energy (kcal) was calculated according to Equation (2). All chemical analyses were performed in duplicates.
Total carbohydrates = 100 − (g of moisture + g of protein + g of fat + g of ash + g fibre)(1)
Energy = 4 × (g of protein + g of carbohydrates) + 9 × (g of fat) + 2 × (g of fibre),(2)

### 2.5. Volume Analysis of Sponge Cake

The sponge cake samples volume was determined using laser measuring device VolScan Profiler 600 (Stable Micro Systems, Godalming, England, UK) in triplicates. Laser step was set to 1 mm and rotation speed was set to 1 rps. Specific volume was calculated as the volume to mass ratio and expressed in cm^3^/g.

### 2.6. Colour Measurement of Sponge Cake

The colour of crust and cross-section of sponge cakes was determined by colourimeter Konica Minolta CR400 (Konica Minolta Co., Osaka, Japan) in five repetitions per batch. The obtained results were expressed in terms of L* (brightness/darkness), a* (redness/greenness) and b* (yellowness/blueness) according to CIELab system of colours. Browning index (BI) was calculated from equations [25] only for sponge cake crust:BI = [100 × (X − 0.31)]/0.172, where:(3)
X = (a* + 1.75L*)/(5.645L* + a* − 3.012b*).(4)

### 2.7. Textural Analysis of Sponge Cake

The TA XT2 Texture Analyser (Stable Micro Systems, Godalming, UK) equipped with a 30-kg load cell was used to analyse texture of baked sponge cakes. Instrumental settings for TPA test (TPA.PRJ) were taken from the software package (Texture Exponent Software TEE32, version 6.1.6.0. Stable Micro Systems, Godalming, UK). Samples from the centre of the crumb slices were cut into cylinders (35 mm diameter, 12.5 mm thick) and compressed. Measurements were conducted in five repetitions per batch.

### 2.8. Sensory Analysis of Sponge Cake

Descriptive sensory analysis was performed with a panel of 10 well-trained sensory panellists (seven women and three men, median age of 33 years) employed at the Institute of Food Technology, University of Novi Sad, Serbia. Training of panellists and descriptive terminology creation were performed on the four selected commercial sponge cakes. All panellists agreed on sensory attributes and definitions. The final list consisted of three appearance attributes (shape irregularity, density, cross-section pore uniformity), two odour attributes (overall odour intensity, egg odour), three taste attributes (sweetness, bitterness, sweet aftertaste), one flavour attribute (egg flavour) and four texture attributes (cohesiveness, elasticity, adhesiveness, crumbliness) (Table 2).

The samples were cut in half and every assessor received one half served on plates labelled with 3-digit numbers in sequential monadic manner following a balanced presentation order. Sensory evaluation was performed in two separate sessions, in individual booths at room temperature; tap water was used for palate cleansing. The panellists recorded their perception of attribute intensities on a 10-centimetre continuous linear scale with left side of the scale corresponding to the lowest intensity and the right side corresponding to the highest intensity.

The study was approved by the Ethics Committee of Institute of Food Technology in Novi Sad, University of Novi Sad, Serbia (Ref. No. 175/I/7-3 from 6 July of 2021).

### 2.9. Data Analysis

Measurements were performed in appropriate number of repetitions and data were presented as average ± standard deviation and analysed for the statistical difference by using Analysis of variance (ANOVA) followed by the post-hoc Tukey’s HSD test at *p* < 0.05. Statistical analysis was performed by using software package XLSTAT 2022.1.2. Principal Component Analysis (PCA) was performed on the instrumentally measured textural, colour, rheological and nutritional properties and the sensory attributes scores of the sponge cake samples to differentiate them and to analyse possible relationships between them.

## 3. Results and Discussion

### 3.1. Batter Properties Evaluation

#### Rheological Behaviour

Since product formulation modification in terms of ingredients replacement or addition could affect a production process, batter rheology represents a valuable tool to determine and characterize product performance during different processing operations such as mixing, pouring or baking of batter. The batter for sponge cakes is a complex emulsion and foam system. Sponge cake is a baked product with a porous structure and is classified as a foam-type cake. The ingredients used in sponge cake formulation have a significant impact on the rheological properties of the batter and the texture of the sponge cake [5,14]. The results of obtained rheological properties are presented in Figure 1.

As can be seen from Figure 1, the control sample was characterized by higher storage modulus values than the loss modulus which is related to more pronounced elastic properties over the viscous one. However, the addition of both whey protein and sourdough had a significant influence on the rheology of the batter. The application of both ingredients resulted in lower storage modulus G′ and loss modulus G″ values, as well as higher frequency dependence of the moduli compared to the control sample.

To further determine the viscoelastic behaviour of the samples, the tangent of the phase angle (tan δ) was measured. The phase angle δ (tan δ = G″/G′) was calculated at defined frequency of 1 Hz. A tan δ < 1 value indicates a predominantly elastic behaviour, whereas tan δ > 1 indicates a predominantly viscous behaviour. The control sample expressed predominantly elastic behaviour with the lowest value of tan δ (0.51). Compared to the control, all other samples had higher tan δ values, which means that the addition of whey proteins and sourdough affected viscoelastic properties towards more viscous behaviour. Namely, the sample containing only sourdough (without whey proteins) had predominantly elastic behaviour with slightly higher values of tan δ (0.75) compared to the control. The P_20_ sample was characterized with similar G″ and G′ values (tan δ = 0.97). On the other hand, the samples with the addition of both ingredients expressed higher frequency dependence and pronounced viscous behaviour. Samples S_20_P_20_ and S_30_P_20_ were characterized with tan δ values of 1.08 and 1.21, respectively. Phase angle value further increased in the sample without baking powder (1.30) underling the significance of this leavening agent in cross-linked structure formation.

Although a decrease in batter elasticity with sourdough fermentation as a result of protein degradation and conformational changes due to proteolytic activity and environment acidity is already documented [26], the impact of protein addition on batter rheology widely depends on protein type and cereal used in bakery product development. According to Assad Bustillos et al. and Majzoobi et al. [9,10], flour substitution with pea and soy protein increased batter consistency and storage modulus mostly due to higher water holding capacity and mean particle size and lower solubility of protein compared to flour. The opposite behaviour was observed in the present study. This is in agreement with the fact that whey protein exhibits higher solubility compared to pea and soy protein [27], thus having less pronounced thickening properties. Therefore, the substitution of a part of gluten-containing wheat flour with whey protein led to gluten dilution and protein solubilisation in batter matrix without structure forming and crosslinking effect as evident from phase angle values. Although sourdough addition lowered system elasticity to extend lower than protein incorporation, their mutual addition led to a more pronounced batter structure weakening.

### 3.2. Baked Sponge Cake

#### 3.2.1. Chemical Composition

The chemical composition of prepared sponge cake samples is shown in Table 3. As expected, the proximate composition is highly influenced by used ingredients. The protein content of samples statistically differed from control, where it was evident that the addition of whey protein significantly improved the protein content of prepared samples. The highest values were achieved in sample P_20_ where only whey proteins were added, while the lowest protein content was detected in the sample without whey protein addition S_30_. According to Regulation (EC) No 1924/2006 of the European Parliament and of the council of 20 December 2006 on nutrition and health claims made on foods, samples with whey protein addition (P_20_, S_20_P_20_, S_30_P_20_ and S_30_P_20_BP_0_) could be claimed as a source of protein.

In general, the addition of whey protein to a formulation led to progressive protein increase, and consequent carbohydrate decrease, which is in agreement with the studies performed by Alves et al. [28] and Camargo et al. [29]. Moreover, samples containing whey protein, except the one without baking powder, were characterized by higher ash content. Alp and Bilgiçli [30] and Zavareze et al. [31] have also found an increase in ash with the addition of a protein source when compared to the control cake. Furthermore, Ramya and Anitha [32] found higher moisture content in sponge cakes having high crude fibre content due to fibre water-binding capacity, while Levin et al. [33] reported whey proteins’ water retention capacities, which may explain the higher moisture content in P_20_, S_30_ and S_30_P_20_ samples.

#### 3.2.2. Physical Properties

Table 4 summarizes the values of sponge cake physical properties (mass, height, and specific volume). Regarding the specific volume, as one of the key determinants of sponge cake quality, the incorporation of whey protein and sourdough, individually or combined, influenced the significant decrease in this parameter with this effect being more pronounced for the sample with the addition of proteins solely. The lowest value was observed in the sample without the addition of baking powder (S_30_P_20_BP_0_).

In the sponge cake systems, a limited amount of water is available, creating a competitive atmosphere in formulations with many different ingredients. The amount and composition of the aqueous phase are of high priority for batter density, where the great importance belongs to the surface-active components such as proteins which, together with other surface-active components, form films around trapped air bubbles [34]. Taking into account that whey proteins can form hydrogen bonds with water molecules [35] and that the incorporation of the sourdough introduced somewhat polymerized gluten in the batter system, as well as fibres, it could be assumed that redistribution of water from the gluten to the other constituents caused partial dehydration and/or conformational changes in the gluten network. The newly created protein network might limit the air expansion process causing a decrease in the specific volume of final products. Another reason could be a lower elasticity of the batter prepared with protein and sourdough compared to Control (Figure 1) leading to the inability of the batter to retain gas bubbles formed during baking. These hypotheses should be confirmed by further experiments such as modification of the preparation process in terms of mixing time and greater work input. However, the combination of protein and sourdough led to an improvement in cake volume in comparison to protein addition alone. On the contrary, spontaneously fermented sourdough could not act as the only leavening agent in sponge cake production

As can be seen from the longitudinal and transversal section (Figure 2), the incorporation of ingredients affected structure and gas bubbles distribution in the sponge cake samples. The presence of organic acids in sourdough, which reacted with a leavening agent, resulted in the appearance of large gas bubbles. However, the incorporation of protein contributed to the modification of network structure and the most homogenous layout of gas bubbles were obtained. The combination of protein and sourdough (S_20_P_20_ and S_30_P_20_) resulted in an appealing gas bubbles layout, bearing in mind that good distribution and presence of bubbles influence final product characteristics such as texture, colour and volume [36]. On the other hand, the absence of baking powder in S_30_P_20_BP_0_ resulted in several large, fused bubbles which negatively affected the mentioned characteristics, as can be seen in Table 4 (specific volume and height of samples).

#### 3.2.3. Colour

The colour of the cake surface crust is one of the critical quality characteristics since it directly influences the initial consumer’s acceptance. Changes in sponge cake formulations significantly (*p* < 0.05) influenced the colour of sponge cake crust and cross-section (Table 4). All enriched samples have statistically (*p* < 0.05) darker crust surface and statistically (*p* < 0.05) lighter cross-section (except sample S_30_) compared to the control sample (Control). Similar results were reported by Sahagún et al. [37], who observed that the addition of whey protein in layer cakes gave darker cakes compared to the control sample. Reported colour changes can be attributed to the Maillard reaction which occurs between free amino acids and reducing sugars during the baking [38]. It can be concluded that by increasing the content of proteins, the content of available free amino groups is increasing as well, which results in the darker colour of the sponge cake crust. In addition, crust darkening in sourdough-containing samples could be ascribed to increased amylase activity, which results in maltodextrins, maltose and glucose liberation [39] and availability for Maillard reaction. On the contrary, all whey protein-containing samples exhibited lighter crumb colours. Unlike some previous studies which reported an increase in gluten-free bread lightness with protein addition due to specific volume increase [40], an increase in sponge cake crumb lightness observed in this study was accompanied by the initial colour of whey protein which was significantly lighter compared to wheat flour. The impact of ingredients’ colour on the colour of the cake crumb was also noticed by Ammar et al. [15], who found that rice flour incorporation in cake led to an increase in cake crumb lightness. Regarding a* values, all new formulated samples presented significantly (*p* < 0.05) higher reddish tones (positive a* values) of cake crust and less greenish tones (negative values of a*) of cake crumb than the control sample. No clear trend was observed for crust and crumb yellowness.

The browning index (BI) indicates the purity of the brown colour. It brings valuable insights into processes in which enzymatic or non-enzymatic browning reactions take place. All enriched sponge cake samples were browner with higher values of browning index.

#### 3.2.4. Texture

The textural properties of prepared sponge cake samples, presented as hardness, springiness, cohesiveness, gumminess, chewiness, and resilience, are given in Table 5. The addition of whey protein and sourdough in the sponge cake formulation containing baking powder significantly (*p* < 0.05) increased the intensities of all analysed textural properties in comparison to the control sample (Control).

The observed correlation between the specific volume and cake hardness is similar to other studies [37], according to which lower specific volume results in higher cake hardness. Increased hardness was observed in all samples containing whey protein and sourdough, with samples containing sourdough without whey protein and baking powder exhibiting the highest hardness. The effect of whey protein on cake texture can be explained by the high solubility of whey protein, which reduces the content of available water necessary for dissolving sugar, which then crystallizes during cake baking, affecting the texture of the final product [14]—bearing in mind that formulations with sourdough were corrected for water content this additionally reduced water available to the ingredients. Furthermore, CO_2_ formed through the chemical reaction between leavening agents such as sodium bicarbonate and lactic acid influenced the expansion of the initially incorporated bubbles during mixing [41]. The size and distribution of the bubbles indirectly affect other quality characteristics. The less aerated structure of the S_30_P_20_BP_0_ sample provides more resistance to compression leading not only to the increased hardness but also to the increased gumminess and chewiness. On the contrary, the S_30_P_20_BP_0_ sample showed the lowest springiness and cohesiveness. Namely, springiness is a textural parameter that indicates the ability of the sample to recover its height between two compression cycles in the TPA test. High springiness values are associated with a fresh, aerated product [42]. The addition of whey protein and sourdough in formulation significantly (*p* < 0.05) increased springiness of samples but only if leavening agent was added in formulation (P_20_ < S_20_P_20_ < S_30_P_20_ < S_30_ < Control). The sample containing whey protein and sourdough without a leavening agent (S_30_P_20_BP_0_) was significantly (*p* < 0.05) denser and had a lower ability to respond to deformation. Cohesiveness is a texture parameter that indicates the energy required for the second compression in the TPA test. This parameter can be correlated with the crumbliness, denseness and chewiness of products perceived by human senses [42].

#### 3.2.5. Sensory Evaluation

The results of the descriptive sensory analysis are presented in Table 6.

All samples containing protein and/or sourdough created a more cohesive, denser, and less crumbly and adhesive structure compared with the control muffin. The less aerated structure of these products affected their odour and flavour. These samples showed a less pronounced overall odour, odour on eggs and flavour on eggs. According to Sahin et al. [43], lower CO_2_ production affects not only the texture but also the aroma of the final product. Furthermore, the incorporation of protein and sourdough caused an increase in sweet taste perception; however, only when 30% sourdough was added in formulations containing baking powder (S_30_ and S_30_P_20_) was the increase in noticed sweetness significant (*p* < 0.05). Accordingly, bitterness detected in the control sample was dramatically reduced in all other samples. These results may be the consequence of biosynthesis of the low-calorie sugar alcohols by lactic acid bacteria and yeasts in sourdough. Sahin et al. [43] discussed in detail the production of different polyols by naturally present microorganisms in sourdough systems.

The absence of sodium bicarbonate in sponge cake formulation had a detrimental effect mostly on the textural properties producing a poorly aerated structure with a firm, dense, compact crumb structure that is less elastic. Similar results were observed with the instrumental measurements of textural properties. The main mechanisms that lead to foam destabilization are drainage-driven by gravity, coarsening due to pressure differences between bubbles of different sizes and coalescence, which is the bursting of liquid films separating neighbouring bubbles.

Principal Component Analysis (PCA) was performed to explore the relationship between sensory attributes and instrumentally measured textural, colour, rheological and nutritional properties of sponge cake samples (Figure 3). PCA biplots clearly show a distinction between Control, S_30_ and S_30_P_20_BP_0_ samples in different quadrants, while the separation of the other three samples was not so clear, which indicates that they are moderate in most of the analysed parameters. S_30_P_20_BP_0_ sample is separated mostly based on significantly higher values of instrumentally measured hardness, gumminess and chewiness, and high content of proteins. Contrary to these, the control sample is mostly separated due to highly pronounced crumbliness, intense overall odour, odour and flavour of eggs, and persistent intense sweet aftertaste evaluated by a sensory panel. Looking at the PC1–PC3 factorial plane, it can be observed similarities between the control and S_30_ in terms of high bitterness, sweetness, adhesiveness, pore uniformity and shape irregularity, high content of carbohydrates and low content of proteins. The obtained biplots emphasize the predominance of whey proteins in the determination of almost all quality parameters.

## 4. Conclusions

The results of this study have shown that the replacement of flour with whey protein concentrate, as well as spontaneously fermented sourdough, led to the final product with specific quality characteristics. These ingredients significantly influenced batter viscoelastic properties, moving from predominantly elastic behaviour in the control sample to predominantly viscous behaviour in a combination of sourdough and whey protein concentrate. Both added components with their constituents, such as simple sugars and different types of proteins, resulted in darker crust colours (due to Maillard reactions) and changes in sensory (appearance, odour, flavour, taste and texture) and textural properties (denser structure with lower specific volume). Despite the observed deteriorated textural properties, the overall characteristics of the final product were at a satisfactory level. In addition to the improvement in the nutritional quality of formulated sponge cakes, the results of this study suggest that sourdough addition can contribute to the improved volume and appearance of protein-enriched sponge cakes. Further research should be focused on the modification of sponge cake formulation with tested raw materials as well as the exploitation of the potential of other protein and sourdough sources in order to improve product techno-functionality, while preserving its nutritional potential.

## Figures and Tables

**Figure 1 foods-11-01992-f001:**
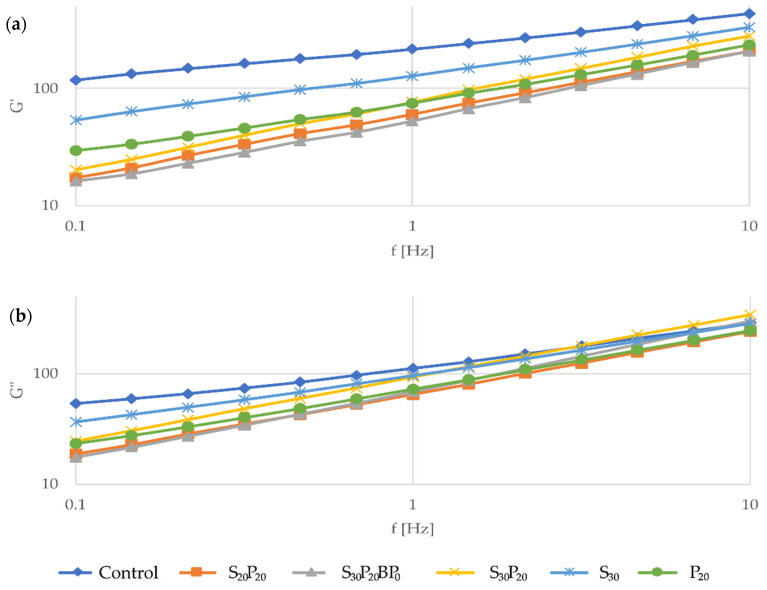
Rheological properties of sponge cake batter samples, dependence of storage and loss modulus in regard to frequency: (**a**) Storage modulus G′; (**b**) Loss modulus G″.

**Figure 2 foods-11-01992-f002:**
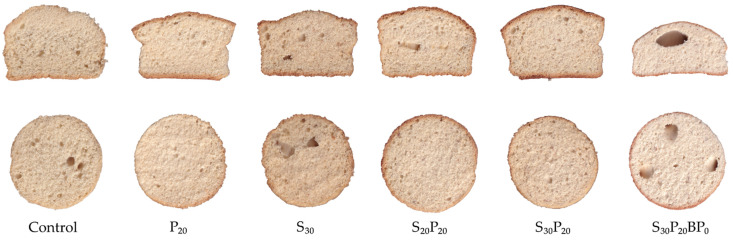
Sponge cake samples longitudinal (**top**) and transversal (**bottom**) section.

**Figure 3 foods-11-01992-f003:**
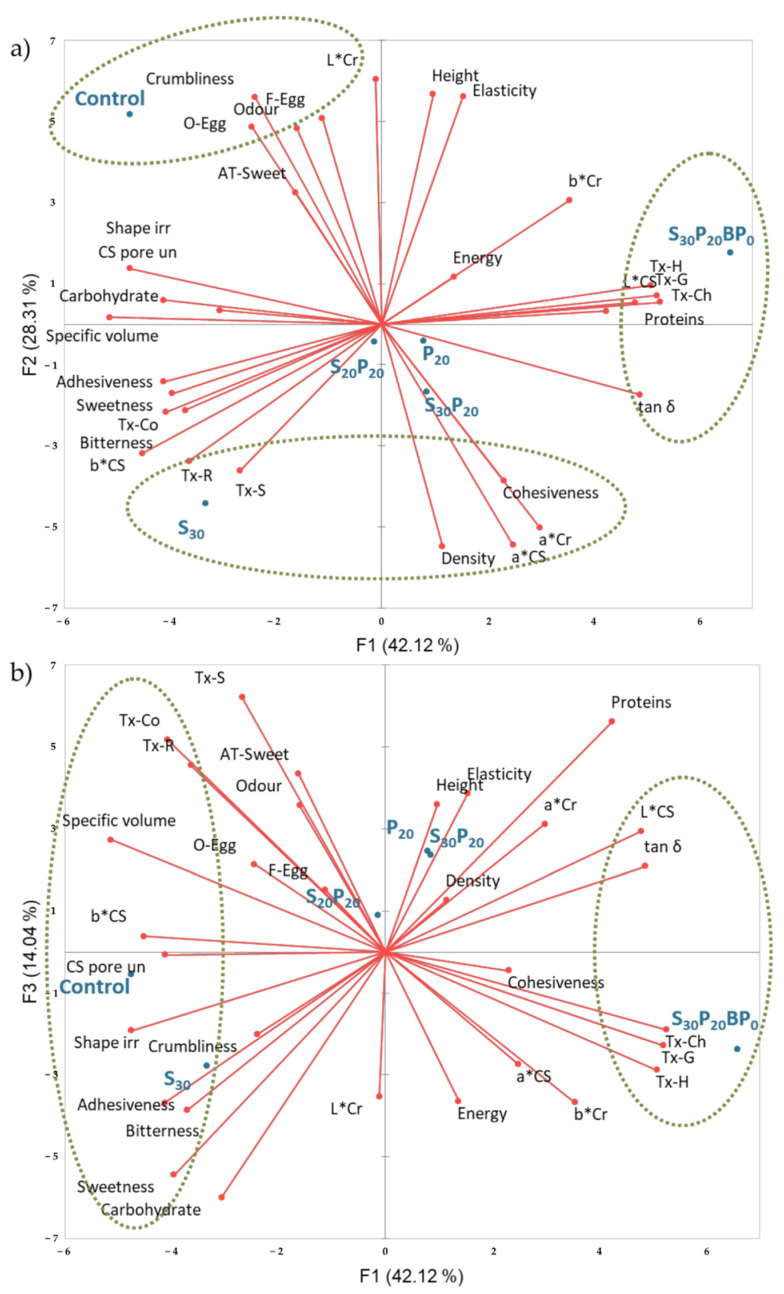
Principal Component Analysis (PCA) biplots representing the six sponge cake samples and sensory and instrumentally measured textural, colour, rheological and nutritional properties in (**a**) F1 and F2 factorial plane and (**b**) F1 and F3 factorial plane. CS pore un—Cross-section pore uniformity; Shape irr—Shape irregularity; O-Egg—Egg odour; F-Egg—Egg flavour; AT Sweet—Sweet aftertaste; Instrumentally measured textural parameters in abbreviations are indicated with Tx: H—Hardness, S—Springiness, Co—Cohesiveness, G—Gumminess, Ch—Chewiness, R—Resilience.

**Table 1 foods-11-01992-t001:** List of the ingredients and formulation of sponge cake samples.

Ingredients (g)	Control	P_20_	S_30_	S_20_P_20_	S_30_P_20_	S_30_P_20_BP_0_
Wheat flour	100	80	85	70	65	65
Whey protein (P)	-	20	-	20	20	20
Sourdough (S)	-	-	30	20	30	30
Sucrose	85	85	85	85	85	85
Egg albumen	52.5	52.5	52.5	52.5	52.5	52.5
Egg yolk	32.5	32.5	32.5	32.5	32.5	32.5
Water	35	35	20	25	20	20
Baking powder (BP)	3.5	3.5	3.5	3.5	3.5	-

**Table 2 foods-11-01992-t002:** Sensory attributes and definitions.

Sensory Attribute	Definitions
Shape irregularity	Degree to which the shape of the sample deviates from the defined one. (Low–Enormous)
Denseness	The number of air holes incorporated in the crumb. (Airy–Densely)
Cross-section pore uniformity	Shape and size homogeneity of the holes in the crumb. (Uniform–Uneven)
Overall odour intensity	Intensity of overall odour of the sample perceived by direct sniffing of the sample. (Low–Intense)
Egg odour	The intensity of odour typical of scrambled egg perceived by direct sniffing of the sample. (Low–Intense)
Sweetness	Perception of characteristic taste of sugar. (Low–Intense)
Bitterness	Perception of characteristic taste of coffee. (Low–Intense)
Sweet aftertaste	Degree of sweet taste intensity perceived after chewing the sample. (Low–Intense)
Egg flavour	The intensity of odour typical of scrambled egg perceived during chewing of the sample. (Low–Intense)
Cohesiveness in mass	Degree to which the chewed sample holds together. (Loose–Compact)
Elasticity	Ability of the product to return to the starting position after tactile compression. (Plastic–Elastic)
Adhesiveness	Degree to which the product sticks to the palate after compression between the tongue and the palate. (Low–Intense)
Crumbliness	Degree to which sample disintegrates or breaks down after compression. (Low–Intense)

**Table 3 foods-11-01992-t003:** Chemical composition of sponge cake samples (g/100 g).

	Control	P_20_	S_30_	S_20_P_20_	S_30_P_20_	S_30_P_20_BP_0_
Moisture	23.37 ± 0.06 ^d^	24.35 ± 0.04 ^b^	23.63 ± 0.07 ^c^	23.38 ± 0.01 ^d^	24.72 ± 0.02 ^a^	22.11 ± 0.05 ^e^
Proteins	9.14 ± 0.02 ^e^	14.79 ± 0.06 ^a^	7.94 ± 0.03 ^f^	13.45 ± 0.07 ^d^	13.74 ± 0.11 ^c^	14.19 ± 0.02 ^b^
Fat	3.86 ± 0.09 ^b^	2.91 ± 0.04 ^d^	3.88 ± 0.08 ^ab^	3.53 ± 0.01 ^c^	3.95 ± 0.08 ^ab^	4.05 ± 0.07 ^a^
Ash	1.28 ± 0.01 ^c^	1.35 ± 0.01 ^b^	1.28 ± 0.00 ^c^	1.18 ± 0.00 ^d^	1.44 ± 0.00 ^a^	0.70 ± 0.00 ^e^
Carbohydrates	59.84 ± 0.04 ^b^	54.58 ± 0.13 ^e^	59.64 ± 0.04 ^a^	55.49 ± 0.07 ^d^	53.02 ± 0.21 ^e^	55.80 ± 0.13 ^c^
Total fibre	2.51 ± 0.07 ^c^	2.02 ± 0.04 ^d^	3.63 ± 0.09 ^a^	2.77 ± 0.05 ^c^	3.13 ± 0.04 ^b^	3.15 ± 0.06 ^b^
Energy (kcal)	315.7	307.7	312.5	313.1	308.8	322.7

Values in the same row marked with different small letters in superscript are statistically different (*p* < 0.05).

**Table 4 foods-11-01992-t004:** Physical properties and colour of sponge cake samples.

	Control	P_20_	S_30_	S_20_P_20_	S_30_P_20_	S_30_P_20_BP_0_
Mass (g)	24.47 ± 0.61 ^c^	26.58 ± 0.15 ^a^	25.39 ± 0.31 ^b^	25.71 ± 0.17 ^b^	26.61 ± 0.30 ^a^	25.64 ± 0.09 ^b^
Height (mm)	42.66 ± 1.55 ^a^	38.11 ± 0.62 ^c^	39.95 ± 2.51 ^c^	40.01 ± 2.55 ^b^	39.53 ± 0.59 ^bc^	31.52 ± 0.96 ^d^
Specific volume (mL/g)	3.16 ± 0.02 ^a^	2.68 ± 0.03 ^c^	2.86 ± 0.08 ^b^	2.80 ± 0.03 ^bc^	2.80 ± 0.02 ^bc^	1.96 ± 0.09 ^d^
Colour—crust						
L*	77.59 ± 1.78 ^a^	62.62 ± 3.52 ^c^	59.25 ± 6.66 ^d^	63.22 ± 5.13 ^c^	56.72 ± 3.04 ^d^	71.99 ± 2.53 ^b^
a*	3.67 ± 1.57 ^d^	13.96 ± 1.23 ^b^	12.49 ± 2.91 ^c^	13.53 ± 1.60 ^bc^	15.31 ± 0.90 ^a^	12.52 ± 2.25 ^c^
b*	36.22 ± 1.19 ^c^	37.07 ± 0.96 ^b^	34.98 ± 1.11 ^d^	34.91 ± 1.98 ^d^	34.08 ± 1.05 ^d^	39.87 ± 0.98 ^a^
BI	63.57	100.6	99.36	91.90	105.84	89.24
Colour—cross-section						
L*	77.42 ± 1.31 ^d^	83.32 ± 0.68 ^b^	76.90 ± 0.81 ^d^	83.23 ± 1.12 ^b^	81.15 ± 0.80 ^c^	84.96 ± 0.70 ^a^
a*	−1.82 ± 0.18 ^d^	−0.68 ± 0.24 ^c^	0.34 ± 0.22 ^a^	−0.65 ± 0.14 ^c^	−0.14 ± 0.20 ^b^	−0.01 ± 0.17 ^b^
b*	25.33 ± 1.10 ^b^	25.23 ± 0.81 ^b^	28.49 ± 0.80 ^a^	23.49 ± 0.77 ^c^	24.92 ± 0.77 ^b^	19.87 ± 0.37 ^d^

Values in the same row marked with different small letters in superscript are statistically different (*p* < 0.05). L*—colour lightness, a*—redness/greenness, b*—yellowness/blueness, BI—browning index.

**Table 5 foods-11-01992-t005:** Textural properties of sponge cake samples.

	Control	P_20_	S_30_	S_20_P_20_	S_30_P_20_	S_30_P_20_BP_0_
Hardness	23.71 ± 1.14 ^d^	62.86 ± 6.01 ^b^	36.67 ± 2.67 ^c^	69.83 ± 9.52 ^b^	67.33 ± 5.87 ^b^	206.18 ± 4.91 ^a^
Springiness	0.92 ± 0.01 ^b^	0.96 ± 0.01 ^a^	0.95 ± 0.01 ^a^	0.96 ± 0.02 ^a^	0.96 ± 0.01 ^a^	0.88 ± 0.00 ^c^
Cohesiveness	0.79 ± 0.01 ^c^	0.81 ± 0.01 ^a^	0.80 ± 0.00 ^ab^	0.80 ± 0.01 ^ab^	0.80 ± 0.00 ^bc^	0.66 ± 0.00 ^d^
Gumminess	18.68 ± 0.99 ^d^	50.79 ± 4.54 ^b^	29.28 ± 2.08 ^c^	55.96 ± 7.80 ^b^	53.67 ± 4.46 ^b^	136.89 ± 2.38 ^a^
Chewiness	17.20 ± 1.11 ^d^	48.80 ± 4.55 ^b^	27.78 ± 2.19 ^c^	53.39 ± 6.17 ^b^	51.49 ± 3.62 ^b^	119.76 ± 2.40 ^a^
Resilience	0.32 ± 0.02 ^b^	0.36 ± 0.01 ^a^	0.36 ± 0.01 ^a^	0.34 ± 0.02 ^ab^	0.34 ± 0.00 ^ab^	0.26 ± 0.00 ^c^

Values in the same row marked with different small letters in superscript are statistically different (*p* < 0.05) according to Tukey’s HSD post-hoc test.

**Table 6 foods-11-01992-t006:** Sensory descriptive analysis of sponge cake samples.

	Control	P_20_	S_30_	S_20_P_20_	S_30_P_20_	S_30_P_20_BP_0_
Appearance						
Shape irregularity	48.5 ± 0.7 ^a^	18.5 ± 4.9 ^c^	38.0 ± 7.1 ^ab^	38.5 ± 14.8 ^ab^	23.5 ± 3.5 ^bc^	14.5 ± 6.3 ^c^
Density	27.0 ± 4.2 ^d^	59.0 ± 1.4 ^abc^	63.5 ± 7.8 ^ab^	68.0 ± 5.7 ^a^	52.5 ± 6.4 ^bc^	48.5 ± 0.7 ^c^
Cross-section pore uniformity	54.5 ± 0.7 ^a^	19.5 ± 13.4 ^b^	43.0 ± 10.6 ^a^	36.0 ± 9.9 ^ab^	48.5 ± 10.6 ^a^	19.5 ± 4.9 ^b^
Odour						
Odour intensity	71.0 ± 9.9 ^a^	62.5 ± 3.5 ^a^	23.0 ± 11.3 ^c^	54.0 ± 11.3 ^ab^	29.5 ± 0.7 ^c^	35.5 ± 13.4 ^bc^
Egg odour	80.0 ± 5.7 ^a^	57.5 ± 3.5 ^a^	16.0 ± 7.1 ^b^	23.0 ± 9.9 ^b^	19.0 ± 2.8 ^b^	21.0 ± 2.2 ^b^
Taste						
Sweetness	47.5 ± 0.7 ^ab^	34.5 ± 7.8 ^bc^	64.0 ± 12.7 ^a^	33.5 ± 2.1 ^bc^	26.5 ± 9.2 ^c^	28.5 ± 10.6 ^bc^
Bitterness	17.5 ± 2.1 ^a^	0.0 ± 0.0 ^c^	33.0 ± 5.7 ^a^	26.5 ± 14.8 ^a^	4.5 ± 0.0 ^b^	0.0 ± 0.0 ^c^
Sweet aftertaste	33.0 ± 11.3 ^a^	11.0 ± 1.41 ^bc^	0.0 ± 0.0 ^c^	39.0 ± 12.7 ^a^	22.5 ± 10.6 ^ab^	6.5 ± 4.9 ^bc^
Flavour						
Egg flavour	27.0 ± 5.7 ^a^	21.5 ± 12.0 ^ab^	7.0 ± 4.2 ^b^	6.0 ± 0.7 ^b^	10.5 ± 3.5 ^ab^	14.0 ± 4.2 ^ab^
Texture						
Cohesiveness	39.5 ± 0.7 ^b^	72.5 ± 3.5 ^a^	71.0 ± 1.4 ^a^	39.5 ± 10.6 ^b^	67.5 ± 6.7 ^a^	68.0 ± 0.0 ^a^
Elasticity	80.0 ± 5.7 ^a^	70.0 ± 5.7 ^ab^	38.5 ± 0.7 ^c^	63.5 ± 9.2 ^b^	70.0 ± 8.5 ^ab^	73.5 ± 3.5 ^ab^
Adhesiveness	17.5 ± 0.7 ^b^	13.5 ± 4.9 ^b^	24.5 ± 4.8 ^a^	6.0 ± 2.2 ^c^	6.0 ± 0.0 ^c^	4.5 ± 0.7 ^c^
Crumbliness	25.0 ± 4.2 ^a^	9.0 ± 1.4 ^b^	9.5 ± 6.4 ^b^	10.0 ± 0.0 ^b^	10.0 ± 2.8 ^b^	12.5 ± 10.6 ^ab^

Values are arithmetic mean ± standard deviation (*N* = 20, 10 sensory assessors in two separated sessions; evaluated in a 100 mm continuous linear scale). Values in the same row marked with different small letters in superscript are statistically different (*p* < 0.05).

## Data Availability

Data is contained within the article.

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
