# Peer review of "Effect of Sourdough and Whey Protein Addition on the Technological and Nutritive Characteristics of Sponge Cake"

_foods, 2022, doi:10.3390/foods11141992_

Round 1
Reviewer 1 Report
The manuscript has investigated the influence of whey proteins and sourdough on sponge cake. Several tests have been performed and the results are very interesting.
Comments:
Lines 58-59: create a complex consortium
Line 84: there is a lack of …
Line 89: please add a “Materials” section and write the brand, company, country, … of the ingredients used in the cake formula, and the chemical reagents.
Lines 100-102: paraphrase
Please write the batter preparation process in detail (the order of mixing the ingredients, mixing time and speed, etc.,). Also, add the size of the molds.
How did you determine the ratio of flour, water, and sourdough in the batters?
Line 116: why you have used “dough”? (usually “batter” is used for cake)
Line 192: delete “sheeting, proofing” these processes are not used for cake batter
Line 193: Sponge cake is …
Line 245: was
Line 304: why the lightness of cakes with whey protein was higher than the control?
Line 339: you have stated that “high content of sugar inhibits activity of yeast present in sourdough and decrease CO2 production.” Do the cake batters have a proofing stage or were baked immediately after preparation?
Reviewer 2 Report
In the chemical composition of the baked sponge cake results&discussion, please compare your results with other authors
Reviewer 3 Report
Manuscript is clear and well written and describes the nutritional, technological and sensory characteristics deriving from the use of whey protein and whole wheat sourdough in the sponge cake. However, some aspects relating to nutritional characteristics should be increased
Line 239: The chemical composition lacks the values of the total dietary fiber which is thus included in the total carbohydrates obtained by subtraction. however, the total fiber does not provide 4 Kcal per gram. Enter the total fiber value or eliminate the calorie content and specify that only some aspects of the nutritional composition have been determined.
